# Quantification of clinically applicable stimulation parameters for precision near-organ neuromodulation of human splenic nerves

Isha Gupta [1✉], Antonino M. Cassará [2], Ilya Tarotin[1], Matteo Donega[1], Jason A. Miranda [1],
David M. Sokal [1], Sebastien Ouchouche[1], Wesley Dopson[1], Paul Matteucci[1], Esra Neufeld [2],
Matthew A. Schiefer[3], Alison Rowles[4], Paul McGill[5], Justin Perkins[6], Nikola Dolezalova [7],
Kourosh Saeb-Parsy [7], Niels Kuster[2,8], Refet Firat Yazicioglu[1], Jason Witherington[1] & Daniel J. Chew [1✉]

Neuromodulation is a new therapeutic pathway to treat inflammatory conditions by modulating the electrical signalling pattern of the autonomic connections to the spleen. However, targeting this sub-division of the nervous system presents specific challenges in translating nerve stimulation parameters. Firstly, autonomic nerves are typically embedded non-uniformly among visceral and connective tissues with complex interfacing requirements. Secondly, these nerves contain axons with populations of varying phenotypes leading to complexities for axon engagement and activation. Thirdly, clinical translational of methodologies attained using preclinical animal models are limited due to heterogeneity of the intra- and inter-species comparative anatomy and physiology. Here we demonstrate how this can be accomplished by the use of in silico modelling of target anatomy, and validation of these estimations through ex vivo human tissue electrophysiology studies. Neuroelectrical models are developed to address the challenges in translation of parameters, which provides strong input criteria for device design and dose selection prior to a first-in-human trial.

[1] Galvani Bioelectronics, Gunnels Wood Road, Stevenage SG1 2NY, UK. [2] Foundation for Research on Information Technologies in Society (IT'IS), Zeughausstrasse 43, 8004 Zurich, Switzerland. [3] SimNeurix, LLC, Gainesville, FL, USA. [4] Non Clinical Safety, Research and Development, GlaxoSmithKline, Park Road, Ware SG12 0DP, UK. [5] Bioimaging, GlaxoSmithKline, Park Road, Ware SG12 0DP, UK. [6] The Royal Veterinary College, Hawkshead Lane, North Mymms, Hatfield AL9 7TA, UK. [7] Department of Surgery, University of Cambridge, and NIHR Cambridge Biomedical Research Centre, Cambridge CB2 0QQ, UK. [8] Swiss Federal Institute of Technology (ETH) Zurich, 8092 Zurich, Switzerland. ✉email: Isha.8.gupta@galvani.bio; Daniel.j.chew@galvani.bio

Neuromodulation using active implantable medical devices is a promising new therapeutic avenue for a number of disease indications focussing on chronic inflammation[1–3]; the premise being to selectively modulate autonomic nerves fascicles supplying internal organs that they innervate. This technique has a great potential for treating various patient populations which are refractory to traditional pharmacological, or biological treatments.

Devices that harness electrical impulses to treat diseases are already widely used, originating with cardiac pacemakers over 50 years ago, and more recently expanding into the central nervous system with deep-brain stimulators and spinal cord stimulators for indications such as Parkinson's disease[4,5] and chronic pain[6–8]. More recently neuromodulation opportunities have leveraged the selectivity and reduced invasiveness of the peripheral nervous system stimulation such as neuromodulation of the sacral nerves for bladder control[9–11]. Over the past twenty years, autonomic neuromodulation of the vagus nerve trunk has opened a treatment opportunity for refractory epilepsy[12,13], depression[14,15], and more recently rheumatoid arthritis (RA) and inflammatory Bowel Disease[5,6,16–19]. In a recent clinical trial in RA patients from the company Setpoint Medical, stimulation of the vagus nerve for up to four times daily inhibited the production of the proinflammatory cytokine tumour necrosis factor alpha (TNFα) and significantly improved RA disease severity[19]. The mechanism of action of this anti-inflammatory activity is proposed to be through targeting of the cholinergic anti-inflammatory pathway[16], involving the efferent vagus connections to the coeliac ganglion, which in turn activates splenic nerves to cause the anti-inflammatory effect with monocytes of the spleen. However, stimulation of the vagus nerve can often result in undesired physiological side effects. Direct vagal innervation of the heart can lead to bradycardia and asystole[20–22] and proximity of motor nerves in the neck can lead to cough, hoarseness, voice alteration and paresthesias[23,24]. Central nervous system side effects including nausea, anxiety, and headache[25,26] are additional risk factors due to afferent pathways to the brainstem. More recently, pre-clinical research is building evidence of expansion for bioelectronic modulation of nerves closer to the target organs that have been shown in areas such as ovarian[27], carotid sinus nerve[28], greater splanchnic nerve[29], and splenic nerves[30,31]. In line with this, Galvani Bioelectronics has recently sponsored a clinical trial to demonstrate the clinical feasibility and safety of splenic neuromodulation during minimally invasive surgery for oesophageal cancer to assess impact on the inflammatory response[32]. During this procedure, a stimulation lead will be placed on the exposed splenic neurovascular bundle and stimulation will be applied to evaluate nerve activation.

Prior to clinical applications in humans, the biological effects of peripheral nerve stimulation on the target region and organs are typically assessed in pre-clinical rodent models. The anatomical sites in these pre-clinical models, however, are several folds smaller than in humans. For example, the diameter of the vagus nerve in rats is a single 0.4 mm fascicle vs a multi-fascicle 2 mm diameter nerve in humans[33]. This provides little insight into appropriate levels of dose, or charge requirements in the clinical setting, let alone the correct anatomical requirements for device design. A large animal model (porcine or canine) is, therefore, often used to verify safety and appropriateness of therapy and implant design. However, even in large animal models, inference and extrapolation are usually necessary to translate parameters to human tissue for regulatory approval. Moreover, while large animal models can replicate some of the anatomical variations also seen in humans, they may fall short of accurately modelling variations in human patients arising from differences in age, body-mass index, natural micro- and macro-anatomical variance,

or co-morbidities. To further de-risk clinical studies and establish more appropriate product designs, it is therefore, valuable to confirm that the stimulation parameters identified in animal models translate safely and efficaciously to humans prior to clinical application[34–36].

Here we applied two complementary approaches to address these challenges in a novel translational anatomy of the human splenic neurovascular bundles to determine the estimations of stimulation parameters for activating the human splenic neurovascular bundle. Recent work has provided evidence that stimulation of the splenic nerve in large animal translational models results in a robust inhibition of induced inflammation [Donegà et al. (manuscript under review)] and is currently being developed as a therapy for humans[32].

Firstly, we used histomorphometric estimations from pig and human splenic tissue to design and build representative neuroelectrical computational models for nerve activation through extra-neurovascular cuff electrodes[37–42]. While computational modelling is relatively recent in the field of neuromodulation, it has already been successfully utilised in several neural engineering applications. For example, a Finite Element Method (FEM) of a compound peripheral nerve describes the dependence of activation threshold on conductivity of perineurium and endoneurium[41,43]. Stimulation parameters on the vagus nerve have also been estimated through computational modelling[43]. In this work, we conducted a simulation study representing approximated nerve behaviour. We used multiple neurostimulation scenarios to determine stimulation current-, charge- and charge-density requirements for nerve recruitment in porcine and human splenic neurovascular bundle (SNVB). We then validated our modelling data by measuring electrophysiological parameters in freshly explanted human splenic neurovascular bundle obtained from organ transplant donors after ethical approval and informed consent. We stimulated the explanted SNVB in an ex vivo electrophysiology preparation using a bipolar cuff electrode and recorded the stimulation-evoked compound action potentials (eCAPs) using downstream hook electrodes. These two approaches have enabled the determination of clinical relevant stimulation parameters for implantable device requirements for human use while refining and confirming the electrode configurations for this therapeutic intervention. Improved characterisation of the relationship between stimulation parameters and nerve recruitment is important for development of neuromodulation therapies[44], to ensure efficacy and safety of proposed stimulation parameters before first-in-human trials[45].

## Results

**Histomorphometric characterisation of human and porcine splenic anatomy.** The first objective of this study was to develop an understanding of human and porcine splenic anatomy and estimate the dimensions of the tissue elements in humans and pigs using histological techniques. For this we obtained six samples of SNVB from both humans and pigs. The pig tissue was obtained from farm pigs (45–60 kg body weight—Large White/ British Landrace cross) in accordance with the UK Animal (Scientific Procedures) Act 1986. Human tissue was obtained from transplant organ donors after ethical approval and informed consent and transported at 4 °C in UW (University of Wisconsin) organ preservation solution. The tissue in both cases was immersed and fixed in 10% neutral buffered formalin (NBF) as soon as possible post-excision. The details of age, gender, body mass index, post-mortem intervals and tissue quality are provided in Supplementary Table 1. The post-mortem interval for human samples was in the range of ~12–40 h.

The samples were divided into sequential blocks for histology (see Fig. 1b, Methods—Histology). The blocks were embedded in

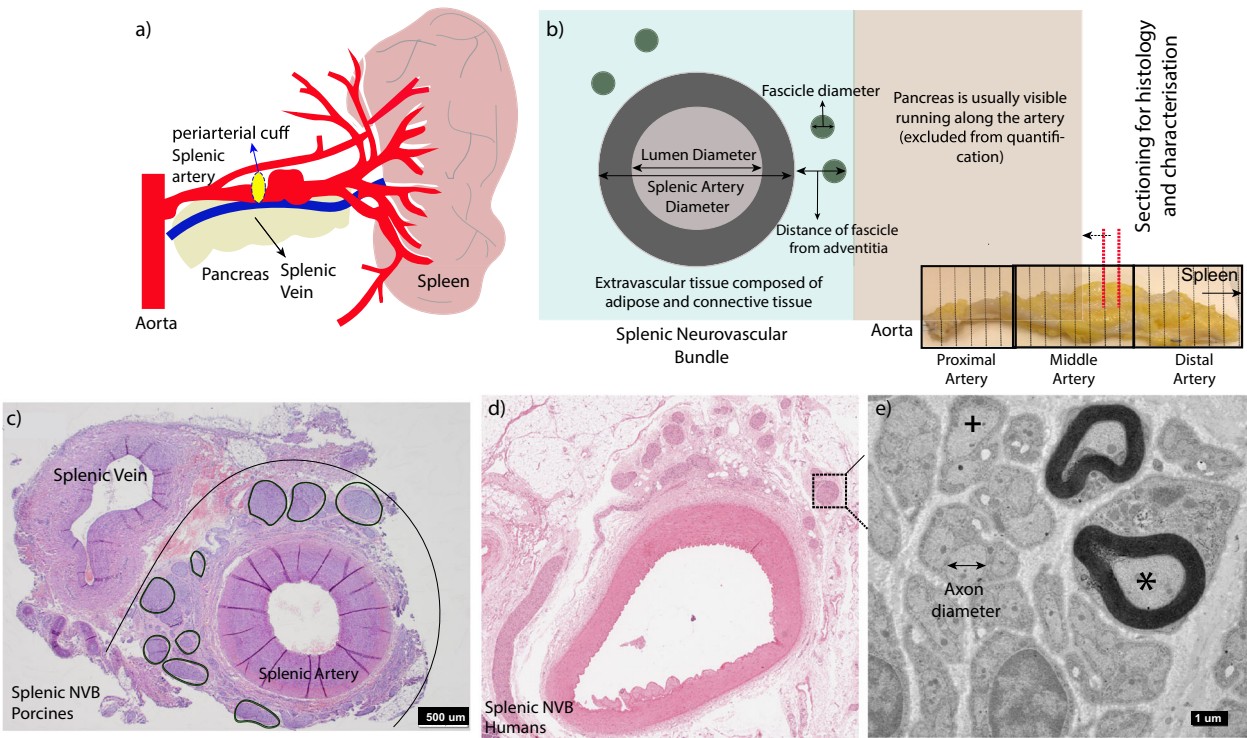

**Fig. 1 Histomorphometric characterisation of human and pig splenic anatomy. a** Anatomical sketch visualising the splenic artery branching from the aorta via coeliac trunk and running to the spleen. A peri-arterial cuff is placed around the mid splenic artery indicated in yellow. **b** Quantification of the histological sections; measurements include distance of the fascicle from adventitia, vessel lumen diameter, splenic artery and fascicle diameter (fascicle diameter was determined using minimum Feret diameter). The sub-image indicates the trimming scheme used to obtain tissue sections of the artery. **c, d** Example of porcine and human H&E histology sections. The U-shaped arc represents the artery and nerves which are placed under the cuff, and black circles highlight individual fascicles. **e** Transmission electron microscopy image of axons within a nerve fascicle used to estimate axon diameter. A myelinated axon is indicated with an asterisk (*); an unmyelinated axon is indicated with an (+).

paraffin, sectioned and stained with haematoxylin and eosin (H&E) (see Fig. 1c, d). All tissue sections were then examined by a histopathologist and glass slides were scanned. Tissue shrinkage due to fixation was roughly estimated at 10–20% based on previous reports in literature[46,47]. The artery diameter is representative of diastolic arterial pressure. Subsequently, the slides were analysed for histomorphometric estimations according to the methodology presented in Fig. 1b. For quantification purposes, the splenic tissue was divided into three parts i.e. proximal (close to aorta), middle (target location for interface) and distal (close to spleen), as shown in Fig. 1b. For splenic neuromodulation, the target location for interface were approximately the middle region of the splenic artery. The quantification for this part is described in the next paragraph for the purpose of building the computational model. In depth characterisation of histomorphometric results including proximal and distal regions are presented in a separate publication [Donegà et al. (manuscript under review)].

In this middle location, for pig, mean Feret fascicle diameter, internal and external splenic artery diameters were estimated to be in the range of approximately 50–250 μm, 0.8–1 mm and 1.5–2.2 mm respectively (see Fig. 1c, Supplementary Fig. 1) [Donegà et al. (manuscript under review)]. In contrast, for humans, middle splenic artery diameter (internal + external) was found to be much bigger, in the range of ~4–6 mm. Fascicle diameter was in the range of 20–400 μm similar to pigs (Fig. 1d, Supplementary Fig. 2).

We next estimated the fascicle spread, i.e. distance away from the tunica adventitia. In pigs, the fascicles were more evenly distributed around the circumference of the artery and in close proximity (<500 μm), whereas in humans, approximately 50% were found in 0–1 mm region, 30% in 1–2 mm, 15% in 2–3 mm and the remaining in about 3–4 mm circumferential region around the artery (Fig. 1c, d, Supplementary Figs. 1, 2). The diameter of the neurovascular bundle (splenic artery diameter + fascicle spread) in humans can thus vary between 6–10 mm approximately (Supplementary Table 2). Substantially more adipose tissue was noted in all the human samples compared to the pig, and some fascicles were also found to be buried in this adipose tissue. It should also be noted that the relative location of fascicles in the loose connective tissue is likely to be affected by post-mortem tissue changes. In particular fat is poorly preserved in formalin-fixed tissues due to dehydration, thus, positions and measurements described herein can only be an approximation of the in vivo scenario.

Finally, a cross section of human SNVB fixed in 10% NBF was examined by Transmission Electron Microscopy (TEM), as shown in Fig. 1e (Methods – Transmission Electron Microscopy). Analysis of images of nerve fascicles showed axon diameters to be in the range of 0.5–1 μm and confirmed the composition to be mainly unmyelinated (Supplementary Fig. 3, approx. >99%). A low magnification image of a large nerve fascicle is also shown in Supplementary Fig. 4 to confirm the composition of the target to be mainly unmyelinated.

**Simulated charge requirements from porcine to human splenic neurovascular bundle.** The electrical stimulation of neurovascular bundles, comprising a loose network of C-fibres represents a novel and significant challenge in bioelectronic medicine. Somatic peripheral nerves contain fascicles that are tightly bound, coalesced, and double-insulated by perineurium and epineurium,

and the axons of interest are typically lower threshold (myelinated A-fibres). Conversely, the bundle of individual fascicles in SNVB are randomly arranged around the splenic artery, surrounded by connective and adipose tissue (varying in pigs and humans), and include fibres that are mostly unmyelinated C-fibres. Evidence of this can be seen in the histology images in Fig. 1, Supplementary Figs. 3 and 4.

Hybrid electromagnetic (EM) and neuronal simulations combining methods of computational electromagnetics and neuroscience are widely used tools used in support of bioelectronic medicine. They were used in this work to predict recruitment of small (0.5 μm) and large (1 μm) unmyelinated C-fibres in two representative image-based and 3D computational neurostimulation models of human and porcine SNVB. Multiple variations of dielectric parameters of the nerve bundles and several stimulus pulse shape (charge balanced biphasic pulses; 0.4 ms, 1 ms or 2 ms per phase) were examined. The two chosen fibre diameters delineate the statistical range of measured C-fibre diameters. Therefore, our modelling work aims at characterising the extreme stimulation conditions instead of a single detailed scenario. The underlying motive of the study was to computationally assess the principal neurostimulation differences between the two species in order to translate the effective stimulation conditions from pig to human for clinical application.

Details of the modelling pipeline are described in the 'Methods' section. Supplementary Fig. 5 also demonstrates a flow chart to describe the stepwise approach followed to build computational simulations. In summary, one representative cross section histological image of SNVBs for each species (see Fig. 2a, b) was segmented using iSEG[48] within the Sim4Life platform[48] (ZMT Zurich MedTech AG, Switzerland). Images of SNVB were differentiated to identify tissue (connective/adipose), vessel wall, blood, extra fascicular medium—internal and external to the electrode—and the endoneurium tissue within fascicles (the associated tissue properties can be found in 'Methods—Dielectric Properties of Tissue'). The segmented tissue surfaces were extruded into 3D structures ('Methods—Computational Model Generation'). The SNVB models were combined with cuff electrode geometries and surrounded by simulated saline solution to mimic experimental conditions. Fascicles were populated with multiple parallel axonal trajectories randomly distributed within each fascicle cross section (Fig. 2, 'Methods—Mesh Generation and EM simulations, Population of Fascicles with Axon Trajectories').

In order to execute neuroelectric simulations the C-fibre Sundt Model[49] ('Methods—C-fibre Electrophysiological Model') was implemented in Sim4Life. The model incorporates the functionalities required to stimulate distributed unmyelinated C-fibres arbitrary fibre diameters. Simulated electric-field distributions, results of the EM simulations, were fed into the C-fibre models scaled by the pulse shape to quantify the dynamic response of the neuronal membrane to the transient stimulation. Ohmic-Current Dominated solver with quasi-static condition was used for all the tissues ('Methods—Dielectric Properties of Tissues'). Titration procedures were then used to quantify pulse-shape dependent stimulation thresholds, as well as strength-duration (S-D), and recruitment curves ('Methods—Coupled EM and Neuronal Simulations and Threshold Current Quantification'). Sensitivity analyses, accounting for instance for variation in dielectric properties of tissues or different pulse parameters were also performed. The creation of neuroelectric models, setup of hybrid EM-neuronal simulations and the post-processing of the results was assisted by Python scripts ('Methods—Data processing, Sensitivity Analysis').

For the porcine model, both a closed and split cuff (2.5 mm diameter, 8 mm length, Cortec GmBH, Germany, Ref: 1041.2180.01 with a contact area of $2 \times 6.5$ mm² [50,51]) were modelled to also cover the in vivo situation where the electrode coverage of the splenic neurovasculature can be around 70% (Supplementary Fig. 6) [Donegà et al. (manuscript under review)]. For the human model, a segmented periarterial cuff was applied (Fig. 2i, 'Methods—Cuff design'). The splits in the cuff model and discontinuous electrode coverage represent the surgical procedure during in vivo scenario.

For the porcine cuff electrode model with connective tissue on the inside and saline on the outside, minimal threshold currents for two pulse widths at two fibre diameters were identified. Threshold currents in response to stimuli with a 0.4 ms pulse width was between 5.5 and 9 mA for 1 and 0.5 μm fibre diameter respectively. At 1 ms pulse width thresholds were 2.3 and 3.6 mA respectively (closed electrode model). By increasing pulse widths, the current needed for activation decreases potentially due to neural accommodation and membrane potential stabilisation induced by lengthy stimuli[52]. Full fascicle excitation was estimated between 66.3 and 34.7 mA for 0.4 ms pulse width and between 21.9 and 11.3 mA for 1 ms pulse width (see Fig. 2k for 0.5 μm results, Supplementary Fig. 7). When comparing porcine in silico predicted (Fig. 2k) and recorded in vivo recruitment curves (Supplementary Figs. 6b and 7) [Donegà et al. (manuscript under review)], a good alignment was found with a less than <10% difference in activation thresholds, supporting the suitability of the FEM approach and appropriateness for use in human electrophysiology modelling.

For the specific human cuff electrode model, the interfascicular tissue was assigned the average dielectric properties of fat and connective tissue with the region external to the electrode assigned to saline dielectric properties (setup FCS). Threshold currents in response to stimuli with a 0.4 s pulse width was between 21.7 and 41 mA for 1 and 0.5 μm fibre diameter respectively. At 1 ms pulse width thresholds were 7.7 and 14.1 mA respectively. 50% fibre recruitment was estimated between 25 and 48.2 mA for 1 ms pulse width and between 73.4 and 143 mA for 0.4 ms pulse width (see Fig. 2l, Supplementary Fig. 8). Thresholds above 200 mA for 100% fibre recruitment were found for all pulse widths and fibre diameters mainly due to dielectric properties around the cuff opening (see Fig. 3, Supplementary Fig. 9). Such hypothetical currents may result in safety and feasibility relevant issues, for example, with electrochemical limits of electrode materials and size of the implantable device.

Additional neurostimulation scenarios for human models were also considered where the dielectric properties of the tissue inside & outside the cuff were varied, as shown in Fig. 2m and Fig. 3. For setup F (Fig. 3a), where only the dielectric properties of fat were present both inside and outside the cuff, thresholds for the largest diameter fibres of 1 μm at a pulse width of 1 ms are 2.8 mA. When both fat and connective tissue were included on the inside and outside the cuff (setup FC, Fig. 3b), this threshold increased to 8.3 mA. With the area outside the cuff set to saline and the inside to fat and connective tissue (setup FCS, Fig. 3c), the threshold was 7.7 mA. These results suggest that the change in conductivity within interfascicular tissue has a larger impact on thresholds than the conductivity of the medium outside the electrodes. Thresholds in setup F were three times lower than Setup FCS whereas Setups FC and FCS only differed by 10%.

The fascicular location with respect to the electrode contacts and the cuff openings for the human model - physical dimensions, and large electrical conductivity of the structures largely affect the electric field distribution and stimulation pattern (Fig. 3 and Supplementary Figs. 9, 10). Electric field distributions quickly drop radially from the internal surface of the electrode towards the centre of the vessel and vary for different composition of tissue dielectric properties. The decrease is

**Fig. 2 Computational model generation for porcine and human SNVB. a**, **b** Histology slice from porcine **a** and human middle splenic artery. **b** Red circle indicates the neurovascular bundle placed inside the cuff electrodes, **c, d** Segmentation of the histology slice for building computational models. **e, f** 2D image of FEM model with 3D view demonstrated in **i**, and side view in **j**. The openings/segmentation in the human cuff can be clearly noted. An example of the clinical cuff is also presented. The length of simulated human and porcine NVB is 39.5 mm and 37.7 mm respectively. **g, h** Meshing for the FEM models. **k** Comparative recruitment curves for pig and human NVB. Pink (red) and blue (green) colour indicates the pulse width of 0.4 ms, and 1 ms respectively in human (porcine) simulations. **l** Comparative results for different axon diameter, 0.5 µm (pink) vs 1 µm (blue). **m** Comparative interfascicular tissue conditions for human simulations. Purple colour indicates fat and connective tissue on the inside, and saline on the outside (FCS). Brown colour indicates fat and connective tissue on inside and outside of the cuff (FC), and grey colour indicates only fat inside and outside of the cuff (F). **n** Comparative charge requirements for monophasic (green), and biphasic pulses (black) for two different axons picked randomly in the model.

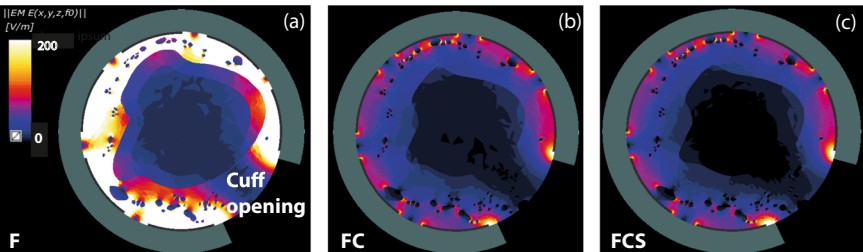

**Fig. 3 Electric field distribution in axial cross section centred on one electrode in the human model.** The different images refer to the different conductivities assigned to the interfascicular and the external tissue. From left to right: **a** Setup F—intra- and extra-fascicular tissue assigned to fat tissue. **b** Setup FC—intra- and extra-fascicular tissue assigned to a mixed composition of fat and connective tissue, and **c** Setup FCS—intra-fascicular tissue associated with a mixed composition of fat and connective tissue, with the saline solution outside.

substantial at the interface with the blood domain because of dielectric discontinuity and within fascicles due to the presence of the large insulating perineurium and large tissue anisotropy. The cuff opening limits the exposure of the fascicles in this region to the main E-field, not on principal current path, and therefore resulting in considerable higher thresholds (Supplementary Figs. 9, 10). Fascicles introduce heterogeneity in the intra-electrode electrical field. Within the fascicles, the electric field is much more homogenous due to shielding by the highly resistive perineurium (Supplementary Fig. 9). A statistical analysis of the activation thresholds within the different fascicles indicates that, in the human model, the recruitment of fascicles in close proximity to the cuff openings (away from the electrical contact pads) requires approximately three times more current than that of fibres in fascicles elsewhere (Supplementary Fig. 10). The relative variability of threshold within fascicles was less than <8%. Furthermore, charge duration curves for biphasic and mono-phasic stimulations were derived for two different randomly chosen fibres as shown in Fig. 2n to theoretically interpret the relationship between 0.4 ms and 1 ms pulse widths. The simulation indicates 1–3 ms as appropriate for biphasic stimulation of nerves in the human model. At such long pulse widths, the charge required for activation for both monophasic and biphasic stimulation is approximately equal[53].

Modelling results suggest that bundle specific characteristics such as the effect of the pulse width, axon diameter, cuff design and interfascicular tissue medium are the principal factors affecting recruitment curves and thresholds. Compared to the effect of current-shunting through the cuff openings and the effect of electrode-fascicle distance, the effect of tissue conductivity outside the electrode is much lower[51]. The results indicate that stimulation thresholds can be reduced by 3 times with 2.5 times increase in pulse width (i.e., 0.4 ms to 1 ms). Thus, the use of wider pulse widths (e.g., >1 ms) for clinical applications can enable reduced current amplitude requirements in IPG design. Current-mode based IPGs are used routinely for spinal cord and deep brain stimulation applications, and provide amplitude ranges of 1–16 mA[54–58]. Leveraging platform

methodology such as described here, early in the research process can provide appropriate system-level requirement definition and integrated circuit (IC) design, including analogue front-end stimulation circuits and power modules.

Furthermore, a number of limitations should be taken into consideration when interpreting the data from the computational models. These include, (i) the uncertainty and variability associated with the dielectric properties of tissues, (ii) potential large anatomical inter-subject variability, (iii) variations in electrode positioning versus fascicle positions and (iv) the use of 2.5D simplified nerve geometry that does not represent the full 3D complexity in vivo. Lastly, this model does not consider the activation of myelinated fibres as the SNVB primarily contains unmyelinated fibres. Myelinated fibres have much lower activation thresholds and are likely to be maximally activated or even blocked at the thresholds of unmyelinated fibres.

**Electrophysiological characterisation of the human SNVB (ex vivo).** The final approach for identifying stimulation parameters in humans was to validate the simulated results using ex vivo human SNVB samples. Figure 4a shows an example of fresh splenic sample from a 63-year-old female donor. The sample was placed in a petri dish, and the SNVB was then carefully surgically-isolated from excess adipose tissue and the splenic vein. A few nerve fascicles were carefully isolated distally for the purpose of recording eCAPs. An isolated fascicle was used as a control and implanted with a smaller diameter cuff electrode (500 μm diameter) for recording and stimulation (Fig. 4b(II)). A larger diameter periarterial cuff of approximately 6 mm diameter was placed arounds the neurovascular bundle (Fig. 4b(I)). Subsequently the tissue with the cuffs was moved into the recording chamber containing circulated, oxygenated Kreb's solution 34–36 °C. The stimulation cuffs were connected to a constant current stimulator (DS5, Digitimer, UK) and the recording hook electrodes were connected to a bioamplifier (BMA-400, CWE, USA) (Fig. 4c, d). For stimulation, a bipolar configuration was used to deliver monophasic pulses at 1 Hz. The neural signal was

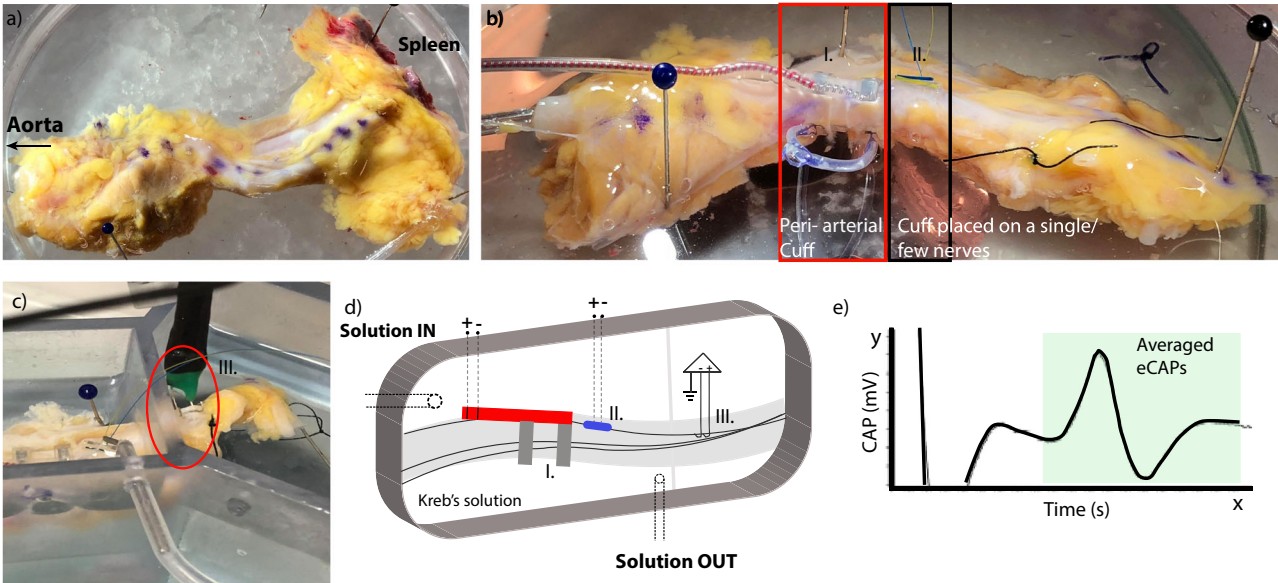

**Fig. 4 Human ex vivo electrophysiology study setup. a** An example of the human splenic tissue as received. The purple dots on the sample indicates the top part of the splenic artery with aorta towards the left end, and spleen on the right end of the sample (for orientation). **b** Placement of peri-arterial cuff around the neurovascular bundle (I), placement of a smaller diameter cuff around few nerves (III). This nerve is dissected, placed in a bath with Kreb's solution, and traced all along till the end of the sample, where the hooks are placed to record compound action potentials **c** (III), **d** Conceptual sketch of tissue with the cuff, and hook placement, and **e** an example of eCAP observed on the oscilloscope.

digitised through an ADC (Power-1401, CED, UK), stored on a desktop computer and visualised on an oscilloscope (Fig. 4e). Activation threshold was defined as the lowest stimulation level that evoked a repeatedly visible eCAP above the background noise in the recording. Maximum axon recruitment was assumed at the point where the maximum eCAP was obtained.

The control isolated splenic fascicle demonstrated a mean threshold of 1.5 mA with charge of 0.6 μC (0.4 ms pulse width), and 100% recruitment at approximately 5 mA with charge of 2 μC (Supplementary Fig. 11). The mean activation thresholds for pulse widths of 0.4 ms, 1 ms and 2 ms are approximately 25.6 ± 8.23 mA, 12.9 ± 4.29 mA and 7.4 ± 3.44 mA respectively (Fig. 5a). The use of a 2 ms pulse width reduces the threshold by 2.5–3-fold, for a 2.5-fold increase in pulse width from 0.4 ms to 2 ms. The demographics of the stimulated human samples are presented in Supplementary Table 3.

Nerve recruitment curves from individual donor samples at different pulse widths of 0.4 ms, 1 ms and 2 ms are illustrated in Fig. 5c, d and e respectively. Due to a maximum current output of the External Pulse Generator (EPG) of 50 mA it was not possible to reach 100% recruitment of nerves using a 0.4 ms pulse width (as seen in Fig. 5c). Increasing the pulse width to 1 ms and 2 ms pulse width was more favourable as a trade-off for reducing current amplitude to complement the IPG output capabilities. Estimated charge density requirements in human ex vivo samples for 100% recruitment is <300 μC/cm²/phase (assuming 18 mm² electrode surface area in this instance).

Importantly, the data obtained experimentally with the ex vivo human tissue is consistent with the computational models. The variation in threshold between the two approaches is approximately 15% for 1 ms and 2 ms and <20% for 0.4 ms pulse widths (see Supplementary Table 4) validating the built human case computational model. It is important to note that the maximum amplitude of the eCAP may only approximately reflect the maximum recruitment of all the axons because the ex vivo measurements are heavily dependent on electrophysiology method accuracy (signal to noise), whereas accuracy in the computational modelling is dependent on accuracy of input criteria. Despite these limitations, these platforms are widely used approaches to evaluate nerve target engagement.

**Simulated chronic charge requirements**. Another significant factor to consider while estimating the charge requirements in a translational setting, is the impact of scar tissue formation (encapsulation) over time in the electrical model. To study the impact of fibrotic tissue ingrowth, and thereby mimicking a chronic therapeutic implantation, we modified the computational model to investigate the effect of scar tissue formation on recruitment curves, stimulation thresholds and charge requirements. In this simulation we used FCS setup with 1 μm fibre diameters. A 0.5 mm layer of scar tissue was positioned surrounding the cuff, both within and outside of the cuff[59]. (Fig. 6, Supplementary Fig. 12, 'Method' – Inclusion of Scar Tissue Development). Charge requirements increased ~1.5 fold from an

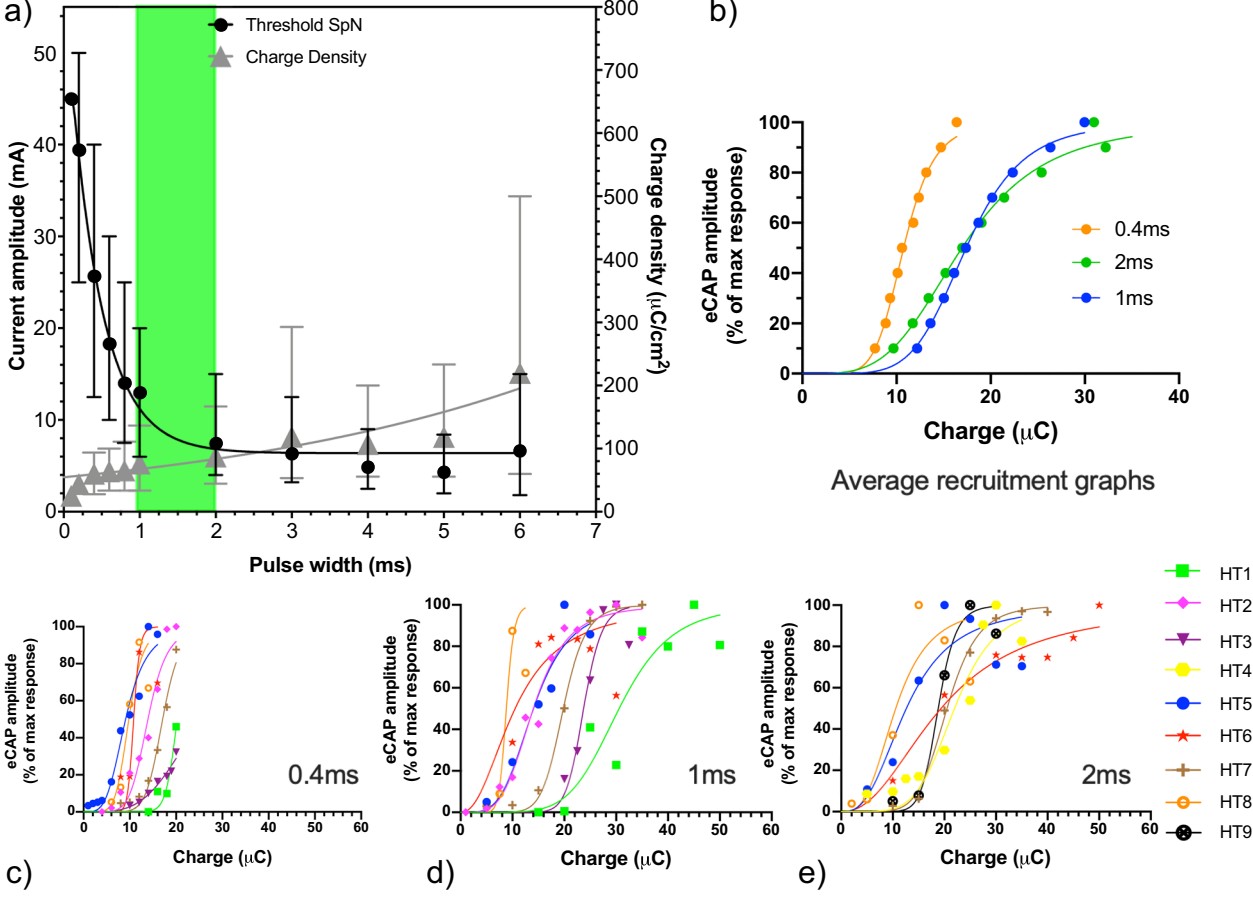

**Fig. 5 Results from ex-vivo electrophysiological study of the human splenic samples. a** Amplitude-Pulse width and Charge Density-Pulse width curves. Charge density is estimated on the basis of 18 mm² surface area of the cuff electrodes. **b** Comparative average recruitment curves for all the human sample at three different pulse widths. For 0.4 ms, the average excludes the samples in which 100% recruitment is not achieved. **c–e** Recruitment graphs from 0.4 ms, 1 ms and 2 ms pulse widths. HT, human tissue, N = 9, Error bars represents range.

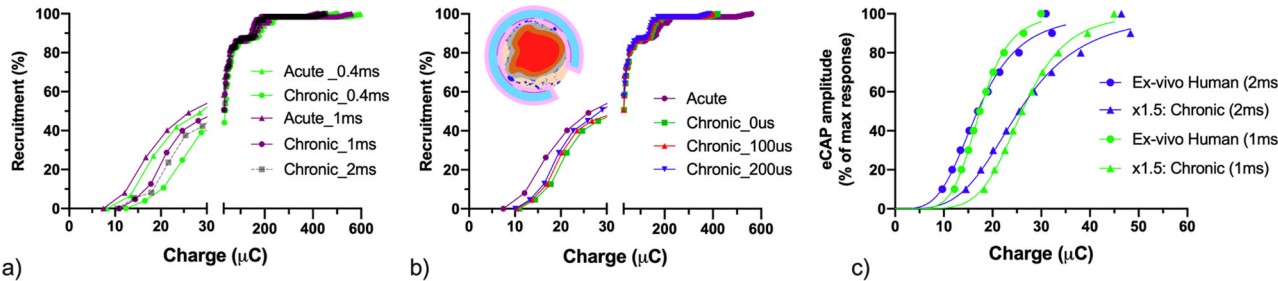

**Fig. 6 Simulated charge requirements in chronic scenario. a, b** Simulated chronic charge requirements in the human computational model with 0.5 mm fibrotic tissue inside and outside of the cuff electrode. Acute and chronic comparison for pulse width 0.4 ms, 1 ms and 2 ms **a**, the effect of interphase delay including 0 ms, 0.1 ms (100 μs) and 0.2 ms (200 μs). **c** A simulated shift in human ex vivo data for 1 ms and 2 ms pulse widths based on the average of the collected ex-vivo data.

acute to chronic encapsulated model for all pulse widths (Fig. 6a). This is based on the assumption of a uniform shape of encapsulation around the electrode and the silicone (Supplementary Fig. 12).

We investigated ways to mitigate the effect of encapsulation by applying longer interphase delay. For a 1 ms pulse width, the acute-chronic increase in charge requirements for 100 μs interphase delay was found to be 47%; whereas increasing the interphase delay to 200 μs mitigates the threshold increase to 35% (Fig. 6b).

Finally, the findings from these simulations were combined with the human ex vivo experimental data (Fig. 5) to understand the approximate charge requirements over time. Applying the 1.5x factor to 1 ms and 2 ms human ex vivo data indicates the requirement of ~50 μC for 100% recruitment of nerves (Fig. 6c). Importantly, the findings from ex vivo and in silico provide guidance on indicative stimulation dose parameters to de-risk clinical trials. The relationship of these findings to in vivo conditions i.e., the relationship between eCAP and organ function, or the therapeutic efficacy will be presented in future publications.

## Discussions

This work provides first account of the neurostimulation parameters to our knowledge effective at activating nerves with the human SNVB with the use of a circumferential cuff electrode. Computational modelling and ex vivo human electrophysiology studies had a significant impact in de-risking the clinical translation path and the definition of user requirements for the IPG. The stimulation threshold in the ex vivo human study for 0.4 ms pulse width was at least three times higher than the in vivo porcine study (approx. 8 mA to 27 mA) [Donegà et al. (manuscript under review)]. As seen in the pig, the stimulation threshold is reduced in human samples when longer pulse widths are used[52]. Longer pulse widths could be considered during IPG and integrated circuit design as a way of achieving activation at lower current amplitudes. Beyond 2-3 ms the charge benefit follows the rule of diminishing returns. Overall factors such as the size of the SNVB, position of fascicles from the metal contact pads, total metal coverage, and importantly cuff architecture affects the electrical recruitment of nerves.

The range from ex vivo human, to their respective in silico studies are found to be aligned within a <15% margin (Supplementary Table 3, and Supplementary Fig. 4). A more detailed physiological comparison between species can be found here [Donegà et al. (manuscript under review)]. These models can be further used to optimise numerous parameters such as cuff designs, placement, IPG design and stimulation capabilities. In this work, modelling analyses were performed for a single porcine and human neurovascular bundle anatomy. To gain a more

comprehensive understanding of the relevant variability of the human models it would be necessary to perform further analysis for additional bundle anatomies. The use of a spatially extended nerve model used in this study does not represent the full 3D anatomical details of nerves, vessels, and surrounding tissues. Fascicles and nerves can separate into lateral branches, include tortuosity and curvature, and have variations in dimensions and spatial distribution along the artery. These factors are not considered in the current models and offer opportunities for refinement in future studies. Moreover, further computational simulations could be performed to assess the demographic heterogeneity, potential dielectric variance of the neurovascular bundle complex and pulsatile nature of the SNVB.

Computational and ex vivo models may also aid in evaluating the likelihood of nerve damage. Nerve damage may result from electric fields or electrochemical processes at the nerve-electrode interface, depending on parameters including charge density[60]. Histomorphometry and electrode design considerations suggest that charge required for nerve activation in this study can be safely applied to the SNVB. For example, distance between the electrode surface and target nerves suggests that damage due to charge density is unlikely to occur. A majority of the nerve fibres in the SNVB are in the mid- to far-field location, particularly after fibrotic encapsulation, reducing the likelihood of nerve damage[60]. Furthermore, the results from this study suggest the upper range charge density required to stimulate the SNVB surpasses the known electrochemical limits of standard smooth platinum iridium electrode[61]. Including high surface area treatments and electrode coatings would increase the limit for stimulation, enabling safer charge transfer to nerves[62–64].

Ultimately these techniques, platforms, and studies are set out to provide confidence in deriving requirement specifications for the neural-electrode interface, and the compliance voltage and output current within the chosen IPG. These have been studied to define stimulation parameters to lead into chronic efficacy studies, and to help refine those prescribed within a first in human intraoperative trial. The use of human tissue for in silico and ex vivo neuromodulation refinement, are recommended as part of the methodological arsenal for the refined design and verification of clinically facing products.

## Methods

### Methods specific to experimental work

*Histology.* Human tissue was obtained from transplant organ donors and transported at 4 °C in UW (University of Wisconsin) organ preservation solution. The tissue was immersed and fixed in 10% neutral buffered formalin (NBF) as soon as possible post-excision. The samples were divided into sequential blocks of 0.5 cm–1.5 cm for histology (see Fig. 1b). The blocks were embedded in paraffin and sectioned at 4–5 μm. The sections were stained with haematoxylin and eosin (H&E). Glass slides were scanned at ×20 using an Aperio Digital Pathology Slide Scanner (Leica Biosystems). The slides were analysed for histomorphometric estimations using NDP viewer as an image analysis tool (Hamamatsu Photonics)[65].

**Table 1 Dielectric properties of tissue.**

| Tissue | $\sigma$ [S/m] | |
|---|---|---|
| | **Human** | **Porcine** |
| Endoneurium (tensor) | $\sigma_{xx} = \sigma_{yy} = 0.166$, $\sigma_{zz} = 0.57$ | |
| Perineurium (thin layer) | $8.7e{-}4m^{-1}$ (*diam[m]) | |
| Silicone | 1e-16 | |
| Connective tissue | 0.38 | |
| Fat | 0.05 | |
| Vessel wall | 0.232 | |
| Blood | 0.666 | |
| Saline solution | 2 | |
| Interfascicular tissue | 0.205 | 0.38 |

*Transmission electron microscopy*. A cross section of human SNVB fixed in 10% NBF was examined by Transmission Electron Microscopy. The tissue, divided into sequential blocks, was post fixed with 1% osmium tetroxide and processed into Agar 100 resin (Agar Scientific Ltd). Resin sections (1 μm thick), stained with toluidine blue, were examined by light microscopy to determine the areas with clear nerve fascicles. Subsequently, 90 nm thick sections were cut, stained with UranyLess (TAAB Laboratories Equipment Ltd) and lead citrate and examined in a Hitachi HT7700 transmission electron microscope. Images of nerve fascicles were taken using Gatan OneView Digital Camera.

*Cuff design*. The cuff electrode is manufactured in silicone and composed of two electrically active arms containing segmented stimulation electrodes and a flexible electrical connection embedded in silicone.

*Electrophysiology characterisation of human SNVB*. Nerve activity in the human electrophysiology study was continuously monitored using an oscilloscope, and digitally recorded with a 1401 acquisition system and Spike 2 v8.0 software with a sample rate set at 20 kHz. Evoked CAPs were averaged (8 pulses) and rectified for quantification of area under the curve of the average response.

### Methods specific to computational modelling

*Image segmentation and modelling electrodes*. A nerve cross section image for each species was chosen from multiple histological images produced for histomorphometric studies. The segmentation of nerve images by reconstructing the edges of the relevant tissues was performed manually within Sim4Life[48] for the porcine model (ZMT Zurich MedTech AG, Zurich, Switzerland). The iSEG modeller was used for automatic recognition of tissues in the human model, in order to identify the principal tissues summarised in Table 1 in the next section with the exclusion of the electrodes and the saline solution. Pre-processing of the image was executed consisting of image-contrast enhancement. The tissue between the fascicles and the electrode was identified as interfascicular tissue with different composition between the two species (connective tissue for the porcine, and mixture of fat and connective tissue for the human). Electrodes were created in Sim4Life using the CAD functionality tools. The thickness of the perineurium was set to 3% of the fascicle diameter in alignment with literature[66].

*Computational model generation*. The Sim4Life platform was used to convert tissue edges from the segmentation into surfaces, to extrude them along the *z*-axis to develop the 3D bundle model, and to create the 3D models of the cuff electrodes. Multiple variations of the SNVBs were initially created with different extrusion length and dimensions of the surrounding saline solution to identify the minimal dimension of the computational domain using EM simulations (see 'Mesh Generation and Electromagnetic Simulation'). This was identified as the volume above which the E-field is reduced from its maximum value (in proximity of the electrodes) below at least −50 db. The criteria were applied to reduce the size of the computational domain without affecting the calculation of the E-field distribution and its influence on the neuroelectric activity of the axons.

*Mesh generation and electromagnetic (EM) simulations*. FEM EM simulations were performed in Sim4Life using the Quasi-Static Ohmic Current Dominated[67] (QS-OCD) Finite Element Method (FEM) solver. It was used to solve the equation $\nabla \sigma \nabla \varphi = 0$, (where σ is the electric conductivity tensor, and φ the electric potential) on adaptive unstructured meshes developed on the model geometries (bundle, electrodes, saline). This solver was chosen as it permits to handle anisotropic electric tensors conductivity and supports thin resistive layer modelling. It is necessary for this study to efficiently model the endoneurium and the perineurium. The use of the Ohmic-Current Dominated solver was justified by the validity of the quasi-static condition $\sigma \gg \varepsilon_0 \varepsilon_r \omega$ (where $\varepsilon_0$ is the electric permeability of vacuum, $\varepsilon_r$ is the relative electric permeability of tissues and ω the angular frequency of the EM field) for all the tissues (see 'Dielectric Properties of Tissues'). This means that system is only purely resistive and that capacitive effects are neglected.

Unstructured meshes mixing tetrahedral and prismatic elements generated with adaptive sizes were built using the 'Multidomain Meshing' tool and optimised for mesh quality. This permitted the fine sampling of the electric field and its gradients even in very small structures (e.g., small fascicles). The meshes were edited to create patches at the interfaces between fascicles and interfascicular tissue and at electrode's surfaces to assign electric flux density boundary conditions. The values were assigned in A/m² permitting an input current of 1 mA between the electrodes. The final human and porcine meshes contained 20.4 M and 31.9 M mesh elements respectively. EM simulations results were executed with high convergence criteria and results were analysed in terms of electric (E-) and current density (J-) field map distributions which were visualised as surface and vector plots.

*Dielectric properties of tissue*. Values of tissue dielectric properties - consisting only the electric conductivity (σ)values - were assigned using the LF-IT'IS database[68]. The conductivity values are reported in Table 1. All tissues and materials were assumed isotropic and homogeneous with the exception of the endoneurium at which anisotropic conductivity values were assigned (longitudinal component parallel to the bundle's axis).

*Population of fascicles with axon trajectories*. Python scripts were created to automize each fascicle with 20 axonal trajectories (lines) extending the whole fascicle's length. This was done by creating a surface triangular mesh on each fascicle's cross section with an average element size of 1/10 of the smallest fascicle's axis and by identifying 20 random (x, y) coordinates of triangle centres as location of the axonal trajectory.

*C-fibre electrophysiological model*. The Sundt[49] electrophysiological model simulating the basic neuro-electric activity of the membranes of unmyelinated C-fibres axons according to the compartmental and cable representation was used in this work. It consists of parameterised values of passive (membrane resistivity and capacity, axial resistance) and active circuital parameters describing the dynamics of axonal membrane. The model was implanted in the Sim4Life platform and its code verified with respect to the original implementation (see ModelDB database, access nr. 187473). The fibre has a specific membrane resistivity (Rm) of 10 kΩ/cm², an axial resistance (Ra) of 100 Ωcm, and voltage-gated Na⁺ and delayed-rectilinear K⁺ channels (KDR) at a density of 40 mS/cm².

*Coupled EM and neuronal simulations and threshold current quantification*. Neuronal electrophysiological simulations were executed for each axonal fibre in the model parameterised as a C-fibre using the Yale's NEURON solver[69] integrated in Sim4Life. The neuro-electric coupling was provided by the established 'extracellular mechanisms' in NEURON that permits the insertion of the extracellular electric potential φ field maps calculated by the EM simulations in the cable equation. Transient simulations were executed in the regime of quasi-static approximation[67] that permits the approximation of the instantaneous electric fields/potentials as the product of E-fields/potential fields map and the (normalised) pulse shape. Simulations provide simulated dynamics of relevant membrane electrophysiological quantities such as the transmembrane electric potential Vm. Simulations were executed with an integration time of 2.5 μs and for a duration of 5 ms, much longer than the largest pulse duration (2 ms) for the sampling of a complete action potential (AP) waveform. The threshold currents identified as the minimal input current eliciting an AP were calculated for each axon in the neurovascular bundle. This was done using an automatic titration procedure based on a bisection algorithm that analyses profiles of transmembrane potentials for different input currents. APs were identified as spikes with depolarisation >80 mV with respect to the resting potential. Recruitment curves were derived from titration tables by cumulative approach.

*Data processing (Python scripts)*. Python scripts were created to facilitate the flexible, parametrised generation of functionalised nerve models, assignment of heterogeneous tissue properties and anisotropic electrical conductivities, creation of mesh and its editing, distribution of fibre models within fascicles, assignment of electrophysiological behaviour and automised post-processing analysis. For example, quantification of stimulation thresholds, extraction of recruitment curves, identify location of spike initiation and latencies (time of first spikes) with respect to stimulus pulse-shape. Python scripts were also created to calculate the surface area of each fascicle's cross section to inversely calculate the diameter of the equivalent circle as $d = 2\sqrt{A}/\pi$.

*Sensitivity analysis*. A sensitivity analysis was executed to verify the impact of the variation the conductivity of the interfascicular tissue on the neuroelectric response of axons, i.e., on location of spike initiation and thresholds. For the human the extreme cases where the interfascicular tissue is associated with only fat ($\sigma = 0.05$ S/m) or only connective tissue ($\sigma = 0.38$ S/m) were considered. For each of this tissue association, EM-neuronal simulations were executed, and thresholds and recruitment curves calculated and compared.

*Inclusion of scar tissue formation*. In order to investigate the neurostimulation properties (thresholds and recruitment curves) of the SNVBs during a hypothetical

chronic implantation stage, the model geometry was modified to include a 0.5 mm thick layer of scar tissue formation around the electrodes and silicone both within and outside of the bundle (see Supplementary Fig. 12). Within the bundle, the scar tissue formation was considered to replace all tissues with exception of the fascicles whose number and positions were unchanged with respect to the acute phase models shown in Fig. 2. Outside of the bundle, the scar tissue was considered to replace a layer of saline solution. The scar tissue was considered isotropic and heterogeneous for both the parts surrounding the electrode and the silicone. Three values of scar tissue conductivities were considered, namely $\sigma_{scar} = 0.15$ S/m, 0.37 S/m (as connective tissue) and 0.56 S/m. Highest value of conductivities were extracted from measured scar tissue conductivities reported in[59].

*Ethical statement.* All human tissue work was conducted in accordance with the UK Human Tissue Authority Code of Practice and Human Tissue Act 2004. Freshly explanted human splenic neurovascular tissue was obtained from organ transplant donors after ethical approval and informed consent from the donor families according to East of England – Cambridgeshire and Hertfordshire Research Ethics Committee reference 16/EE/0227.

Pig tissue was provided by the Royal Veterinary College, Potters Bar, UK. All animal tissue work was conducted in accordance with the Schedule 1 procedures of the UK Animal (Scientific Procedures) Act 1986. The protocol for animal studies was approved by the RVC Animal Welfare and Ethical Review Board and The Galvani Animal and Scientific Review Committee. All animals were housed and transported under conditions specified in the UK's Animal Welfare Act 2006 and The Welfare of Farm Animals (England) Regulations 2007.

**Reporting summary**. Further information on research design is available in the Nature Research Reporting Summary linked to this article.

## Data availability

Source data for Figs. 2, 5 and 6 are provided with the paper. Any other dataset is available from the corresponding author upon reasonable request.

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

## Acknowledgements

The authors thank Karen Cartwright and Nicola McCormack at GlaxoSmithKline (UK) for processing the tissue for histology. We thank Dr. Kourosh Saeb-Parsy, Nikola Dolezalova, Krishnaa Mahbubani at Addenbrooke's hospital (Cambridge, UK) for dissecting and providing valuable human tissue. The authors are also grateful for the generous donation of post-mortem human tissue by fourteen anonymous individuals. We further thank Cindy Au at Verily Life Sciences for providing CAD designs of cuff electrodes for computational modelling.

## Author contributions

I.G., J.W. and D.J.C. designed the study. A.R., I.G., P.G. and K.C. performed the histo-pathological and data analysis studies from human tissue samples. I.G., M.D., D.M.S., W.D. and J.A.M. performed the human ex vivo electrophysiological study. I.G. analysed the data. I.G., A.M.C., M.A.S., I.T., E.N. and N.K. executed and analysed the computational models. S.O., P.M. and R.F.Y. contributed to designing the cuff electrode. K.S.P. and N.D. provided the precious human samples for histology and ex vivo work. J.P. provided valuable pig tissue. I.G., A.C. and D.J.C. wrote the manuscript. All authors contributed in reviewing the manuscript.

## Competing interests

I.G., M.D., D.M.S., J.A.M., D.J.C., S.O., P.M., W.D., R.F.Y. and J.W. are employees of Galvani Bioelectronics and declare that Galvani Bioelectronics has filed international patent applications describing methods, devices, and systems for splenic nerve neuro-modulation. We declare that Galvani Bioelectronics provided funds to support their work associated with this manuscript. A.M.C., E.N., N.K., J.P., K.S.P., N.D. and M.A.S. declare that Galvani Bioelectronics provided funds to support their work associated with this manuscript. The remaining authors declare no competing interests.
