## [Peer Review File · Communications Biology]

Reviewers' comments:

Reviewer #1 (Remarks to the Author):

The work described in the manuscript "Bioelectronic Medicine Translational Development – Quantification of stimulation parameters for precision near-organ neuromodulation of human splenic nerves" by Gupta et al. develops two related approaches toward determining optimal parameters for human peripheral nerve stimulation as a therapeutic tool. This is needed because of the differences between rodent and human nerve morphology. The first approach involves construction of a computational model of nerve activation based on morphology determined from porcine and human tissue. The second method, which is intended to serve as validation of the first, assessed the input/output function based on compound action potentials evoked in an ex-vivo human tissue preparation. The major conclusion is that the input/output relationships compared between the simulations and ex-vivo experimental results are quite close, validating the use of computer simulations as a first step toward determining stimulation parameters for use in humans, and can serve as a means for optimizing the efficacy of peripheral nerve stimulation.

Major Concerns

1. The computational approach used in this study is very difficult to assess. The reference to the 'histomorphometric estimations' and that stimulation of the splenic nerve reduces inflammation has not been published, and therefore cannot be used in the assessment of this approach. Furthermore, the details of the simulations are not described in any detail here, which relies on the Sim4Life platform. Consequently, important aspects of the simulations, such as the spatiotemporal distribution of the electric field cannot be evaluated. Figure 3 is apparently intended to address this question, but there is some confusion as to whether voltage-density is being presented (most likely) or non-uniformities in conductivity. Finally, axonal recruitment will be affected by the impedance properties of the tissue. As a result, variations in stimulation frequency (and number) with simulated pulse trains could potentially influence the efficacy of axonal recruitment.
2. The model characterizes the activation of unmyelinated C-type fibers. However, the authors note on page 8 that the axons of interest are composed of myelinated A-type fibers. The presence of myelination is highly likely to affect the conductivity of the nerve bundle, and therefore the voltage distribution. While in Figure 1, both myelinated and unmyelinated axons are presented as part of the fascicle. It is also not clear how the statistics and dimensions of the fascicles, presented in Supplementary Figure 1, are applied to the model. At least two different axonal diameters are presented in Figure 2I. Likewise, it would be useful to know how the variability in fascicle morphology affects the input/output relationship.
3. Validation of the simulations compared to in vivo porcine and ex vivo human splenic nerve bundle results were consistent in terms of the charge/axon recruitment (input/output) relationship. However, two limitations of this approach are 1) does the maximum amplitude of the compound action potential actually reflect maximum recruitment of all of the axons and 2) how does compound action potential recruitment relate to therapeutic efficacy. As suggested above, the relative efficacy of single bi-phasic pulses (Supplementary Figure 4) could potentially have different effects on recruitment compared with trains of stimuli.

Reviewer #2 (Remarks to the Author):

Gupta et al utilized computational modeling and ex vivo electrophysiology studies to identify the optimal parameters for stimulation of the splenic neurovascular bundle (SNVB). Their approach is

thorough and builds on multiple similar previous efforts which have used a hybrid approach to designing electrical stimulation therapies. Their findings are in line with the existing literature on electrical stimulation and the specific parameters that dictate activation thresholds and recruitment curves.

I have a few follow up questions.

1. The study utilizes complex geometries to determine the optimal stimulation parameters and reports mostly on activation thresholds and recruitment curves. What about charge density at the other tissues in the SNVB during optimal stimulation?

2. In line with #1, one of the primary motives of the study was to understand safety limits, however there is no discussion or mention of how the identified results fit into the current and widely accepted definitions of safety limits for electrical stimulation.

3. The electrode in this study is designed to fit over the splenic artery and the surrounding fat and nerves. In clinical applications, this electrode would be stimulating a structure that is beating and has active blood flow. How might these variations in the state of the tissue impact the efficacy of the stimulation?

4. Figure 4e. What are the values for the axes?

In limitations should also consider application in the intact in vivo condition. There it is much more than profile of axon activation; it is also how the autonomic neural networks respond to such stimuli. For example, using functional organ readouts, thresholds are decreased and response curves left shift when peripheral nerves are disconnected from central aspects. Endogenous reflexes oppose the exogenous bioelectric intervention.

We would like to thank you and the reviewers for the feedback on our manuscript. The constructive comments from the reviewers have allowed us to enhance the quality of our work further.

The following comments have now been full addressed to the best of our knowledge.

Referee expertise:

Referee #1: modeling, electrophysiology

Referee #2: atrial fibrillation

Reviewers' comments:

Reviewer #1 (Remarks to the Author):

The work described in the manuscript “Bioelectronic Medicine Translational Development – Quantification of stimulation parameters for precision near-organ neuromodulation of human splenic nerves” by Gupta et al. develops two related approaches toward determining optimal parameters for human peripheral nerve stimulation as a therapeutic tool. This is needed because of the differences between rodent and human nerve morphology. The first approach involves construction of a computational model of nerve activation based on morphology determined from porcine and human tissue. The second method, which is intended to serve as validation of the first, assessed the input/output function based on compound action potentials evoked in an ex-vivo human tissue preparation. The major conclusion is that the input/output relationships compared between the simulations and ex-vivo experimental results are quite close, validating the use of computer simulations as a first step toward determining stimulation parameters for use in humans, and can serve as a means for optimizing the efficacy of peripheral nerve stimulation.

Major Concerns

1. The computational approach used in this study is very difficult to assess. The reference to the ‘histomorphometric estimations’ and that stimulation of the splenic nerve reduces inflammation has not been published, and therefore cannot be used in the assessment of this approach. Furthermore, the details of the simulations are not described in any detail here, which relies on the Sim4Life platform. Consequently, important aspects of the simulations, such as the spatiotemporal distribution of the electric field cannot be evaluated. Figure 3 is apparently intended to address this question, but there is some confusion as to whether voltage-density is being presented (most likely) or non-uniformities in conductivity. Finally, axonal recruitment will be affected by the impedance properties of the tissue. As a result, variations in stimulation

frequency (and number) with simulated pulse trains could potentially influence the efficacy of
axonal recruitment.

Thank you for this comment. We have worked on this comment part-by part as follows:

(i) *The reference to the ‘histomorphometric estimations’ and that stimulation of the splenic*
*nerve reduces inflammation has not been published, and therefore cannot be used in the*
*assessment of this approach.*

We agree with the reviewer that the ‘*histomorphometric estimations*’ and that
stimulation of the splenic nerve reduces inflammation has not been published yet.
However, this data is in separate stand-alone manuscript that is currently under review,
and for this we have made the following changes in this manuscript and supplementary
material (the changes in the main manuscript and supplementary material are
highlighted in red):

a) **Page 3:** Galvani Bioelectronics has recently sponsored a clinical trial to evaluate the
feasibility and safety of splenic neuromodulation during minimally invasive surgery
for oesophageal cancer. This trial will evaluate the effects on the inflammatory
response as an exploratory read-out. During this procedure, as a primary objective a
stimulation lead will be placed on the exposed splenic neurovascular bundle, and
electrical stimulation will be applied to the lead to test whether the nerves can be
activated in the clinical setting. The details of the clinical trial are published on:
<https://clinicaltrials.gov/ct2/show/NCT04171011> and are now added to the
manuscript to further highlight the growing evidence for development of splenic
neuromodulation to inhibit inflammation. Indeed, the data in this manuscript has
directly led to the parameters of choice in this clinical trial.

b) **Page 4, 5-** The section on histomorphometric estimations has been extended to
include the estimations from pig and human tissue in more detail. Importantly, the
intention of presenting the results in this manuscript is to demonstrate approximate
dimensions of the neurovascular bundle in two species for the purpose of modelling.
The summary of characterisation from the pig and human splenic anatomy in N=6
samples can be found in the histomorphometric section on page 4-5. The
demographics and pathological assessment from each human sample has also been
added to the Supplementary Section (Supplementary Figure 1, Supplementary Table
1, Supplementary Figure 2). The details have been balanced between main
manuscript and supplementary section. If the reviewer suggests, more details can be
moved to the main manuscript. It is also important to note that this manuscript does
not show results or discuss the effects of reduction of inflammation in large animal
models. The main objective of this manuscript is to estimate the stimulation
parameters in humans pre-clinically to de-risk clinical trials. Our work on efficacy in
large animal models i.e. response of pro-inflammatory cytokines in response to

splenic neuromodulation is being reviewed as a separate body of work in PNAS. We
provide you a copy of that manuscript with this letter for your reference.

c) **Page 3:** In terms of stimulation of splenic nerve, we are relying on the preclinical and
clinical references (Koopman et al 2016, Pavlov and Tracey 2012) stating that VNS
leads to a systemic suppression of inflammation via the inflammatory reflex within
the spleen. We are stating that we are accessing the splenic nerve directly and
providing a novel and targeted approach to the spleen. In addition, our attached
manuscript is provided to show the effect of splenic stimulation at inhibiting acute
inflammation to indicate our approach is sound.

(ii) *The computational approach used in this study is very difficult to assess. Furthermore,*
*the details of the simulations are not described in any detail here, which relies on the*
*Sim4Life platform. Consequently, important aspects of the simulations, such as the*
*spatiotemporal distribution of the electric field cannot be evaluated. Figure 3 is*
*apparently intended to address this question, but there is some confusion as to whether*
*voltage-density is being presented (most likely) or non-uniformities in conductivity.*

Thank you for this comment. We have now revised and re-written the entire section
about the computational modelling in the 'Methods' section. To facilitate clarity, we
have firstly divided the Methods section in two parts i.e. 'Methods specific to
Experimental section' and 'Methods specific to Computational Modelling'. On Page 16
of the main manuscript, you will now be able to follow the methodology used for
building and executing the simulations. The methods contain the following subsections:
Image Segmentation and Modelling Electrodes, Computational Model Generation, Mesh
Generation and Electromagnetic Simulations, Dielectric properties of tissue, Population
of Fascicles with Axon Trajectories, C-Fibre Electrophysiological Model, Coupled EM and
Neuronal Simulations and Threshold Current Quantification. Importantly, these
subsections have been revised to add substantial description about how the procedure
was used for the segmentation of the two SNVBs images, about the surface
reconstruction in 2D, extrusion in 3D and creation of a numerical unstructured mesh,
including images. We have added details of the electromagnetic simulations, the theory
behind the use of the quasi-static Finite Element Method (FEM), and the coupling with
the electrophysiological simulations. We have included more details on spatial
distribution of the electric field (EF) in both SNVBs and discussed the impact of
heterogeneity of tissues on the simulated field maps. Secondly, we have added a new
figure in the Supplementary Section (Supplementary Figure 5) which depicts the flow
chart for the computational modelling pipeline and Supplementary Figure 9 which
demonstrates the E-field and Current density maps for two longitudinal cross-sections.

**We hope the reviewer will now find this description appropriate. We are happy to**
**answer or provide more details if the reviewer wishes.**

(iii) *Finally, axonal recruitment will be affected by the impedance properties of the tissue. As*
*a result, variations in stimulation frequency (and number) with simulated pulse trains*
*could potentially influence the efficacy of axonal recruitment.*

This is an interesting point for discussion. The response of body tissues and
neurons/nerves depends on the frequency of the electromagnetic (EM) fields, or the
applied electric currents. The interaction of electric current with the body tissues during
neurostimulation can be broadly categorized in two categories: (a)The (macroscopic)
response of the bulk tissues to applied currents, and (b) the (microscopic/cellular)
electrophysiological response of the neuronal membrane.

(a) Concerning the macroscopic response of bulk tissues, the conductivity of body
tissues is almost constant at low frequencies (intended up to a few tens of kHz). In
addition, the reactive, frequency-dependent component of the Maxwell equation,
proportional to $\omega\epsilon$ (where ω is the angular frequency of the EM field, and ϵ the
relative electric permeability) is negligible at the frequencies of interest. This implies
that the tissues can be considered purely resistive (following the Ohm's law) with
constant conductivity. In our work, we used the ohmic-current dominated (OCD)
unstructured solver that integrates the Maxwell equations in this *quasi-static*
approximation. The use of this approximation is widely accepted in computational
neuroscience [Bossetti CA, Birdno MJ, Grill WM. *Analysis of the quasi-static*
*approximation for calculating potentials generated by neural stimulation. J Neural*
*Eng. 2008;5(1):44–53]. We have now cited this reference in the main manuscript.*

(b) Stimulation pulse shape characteristics such as stimulation frequency and number of
pulses instead affect the electrophysiological response of the axons due to the
electrical properties of the neuronal membrane. The electrophysiological 'Sundt'
model of C-fibre, used in this work, includes passive resistive and capacitive
components calibrated for such fibre type, as well the most relevant
transmembrane ionic channels described as voltage-dependent current sources. The
inclusion of these details permits a realistic modeling of the neuroelectric response
of C-fibres to EM fields. The reviewer is right in saying therefore that the response of
the axon to a single or multiple pulse can be different. However, in this specific
application, pulses will be delivered at inter-pulse distance much longer than the
characteristic time constant of the membrane (approximately 3ms for a C-fibre), i.e.
much longer than the refractory period of the membrane. This means that the
membrane is already at resting state condition before the arrival of the next pulse,
and therefore there is less significance in modeling the response of the axons to the

complete train of stimulation waveform. If the repetition time were to be smaller
than approximately 10ms (100Hz), the response of the bundle to the complete,
complex pulse shape would have been investigated. A discussion about the quasi-
static approximation and its validity is included in the 'Method' 'EM Simulation'
session, while discussion about the transient response of the axon to EM field is
discussed in the 'Method' 'Hybrid EM-Neuronal' section.

We will also cover the effects of frequency to neural activation in the in-vivo setting
in a separate manuscript in future.

2. The model characterizes the activation of unmyelinated C-type fibres. However, the authors
note on page 8 that the axons of interest are composed of myelinated A-type fibres. The
presence of myelination is highly likely to affect the conductivity of the nerve bundle, and
therefore the voltage distribution. While in Figure 1, both myelinated and unmyelinated axons
are presented as part of the fascicle. It is also not clear how the statistics and dimensions of the
fascicles, presented in Supplementary Figure 1, are applied to the model. At least two different
axonal diameters are presented in Figure 2I. Likewise, it would be useful to know how the
variability in fascicle morphology affects the input/output relationship.

Thank you for this comment. We want to begin by clarifying that our target is approximately
>99% unmyelinated. We have now clarified this in main manuscript on Page 6 along with the
statistical values (last paragraph). We have also added another Supplementary Figure 4 as an
example of Transmission Electron Microscopy (TEM) image from a large nerve fascicle. From
this figure, we have quantified the composition of our target. In the main manuscript, we have
attempted to incorporate a high magnification image of a nerve fascicle as an example. If this
leads to confusion, we can eventually remove it. In the computational modelling section, on
Page 8, first paragraph, we have rephrased to highlight that we are only modelling
unmyelinated fibres in order to avoid this misunderstanding. Indeed, it is only the unmyelinated
fibres that are releasing noradrenaline into the spleen, and the ones responsible for any anti-
inflammatory process, so we are mainly interested in this 99%.

In this work, we used tissue conductivity values for the bulk endoneurium considering only
unmyelinated fibres as reported in [*Pelot 2019 - Pelot NA, Behrend CE, Grill WM. 2019. On the*
*parameters used in finite element modeling of compound peripheral nerves. J Neural Eng.*
*16(1):016007*]. The presence of minimal degree of myelinated fibres is unlikely to affect the
conductivity of the nerve bundle, and therefore the voltage distribution. In our case, in fact, it is
less than 1% myelinated fibres which are typically activated at current thresholds much smaller
than for C-fibres. This means that at minimal thresholds for activation of C-fibre, all, or a large
part of the myelinated fibres are expected to be already stimulated (or blocked via conduction
blocking). In this piece of computational modelling work, to reduce the complexity, and
because they are assumed to be irrelevant for the anti-inflammatory effects, we have not
included the 1% myelinated composition. Instead we have discussed this in the first paragraph,

Page 11. Moreover, the activation of such fibres and the targeting of these fibres is not a
priority in this study as the present goal is to determine threshold for a specific treatment.

Furthermore, from the similar TEM image (as can be seen in Supplementary Figure 3), a
quantification study of the range of fibre diameters was performed. The typical range of fibre
diameters for these C- fibres is between 0.5 μ m and 1 μ m. By modelling the response of uniform
distributions of 0.5 μ m or 1 μ m fibres diameters, we aimed at identifying current responses for
extreme scenarios which is now edited in main manuscript, Page 8, paragraph 1. Thresholds are
lower for larger fibres and larger for small fibres. By modelling the response of the two extreme
situations, we identified current responses for two extreme scenarios as shown in Figure 2l. It
would indeed be possible to generate models with a statistical distribution of fibre diameters
(e.g. gaussian, or derived from measurements), however this approach would not provide any
additional information on charge requirements.

Images illustrating the spatial distribution of stimulation thresholds within fascicles were added
in Supplementary Figure 10 for two different pulse amplitudes. The variability and statistics in
the fascicle morphology are also discussed. E-field and current density maps are also now
added in Supplementary Figure 9. These images may clarify some concepts, such as thresholds
for activation of fascicles closed to the cuff opening are large and that the thresholds for
activation of fibres within the fascicles have relatively low variability. Also, Page 10, paragraph 1
summarises our findings. *“A statistical analysis of the intensity and variability of the stimulation
thresholds within the different fascicles indicates that in the human model the recruitment of
fascicles in close proximity to the cuff openings (away from the electrical contact pads) require
three times more current than that of fibres in fascicles elsewhere (Supplementary Figure 10)”*.
The E-field experienced by different axons within the same fascicle is very homogeneous due to
the presence of the largely insulating perineurium.

**3. Validation of the simulations compared to in vivo porcine and ex vivo human splenic nerve**
**bundle results were consistent in terms of the charge/axon recruitment (input/output)**
**relationship. However, two limitations of this approach are 1) does the maximum amplitude of**
**the compound action potential actually reflect maximum recruitment of all of the axons and 2)**
**how does compound action potential recruitment relate to therapeutic efficacy. As suggested**
**above, the relative efficacy of single bi-phasic pulses (Supplementary Figure 4) could potentially**
**have different effects on recruitment compared with trains of stimuli.**

1. The reviewer is right in pointing this. With respect to human ex-vivo data, we indeed missed
to discuss this in the original manuscript. We have now clarified this on Page 11, 12 and 13
in the main manuscript. In this work, we believe that an increase in eCAP peak is due to an
increase in recruitment of number of axons. During the experiment, eCAPs were amplified,
filtered, recorded and observed on the oscilloscope. As you ramp up the stimulation
intensity, the evoked compound action potential peak keeps increasing, representing the
number of axons being recruited. Experimentally, maximum axon recruitment i.e. 100% is
‘assumed’ at the point where the compound action potential is maximum, i.e. the point

where it is visually calculated as a maximum on the oscilloscope. Therefore, the accuracy in
the ex vivo measurement is heavily dependent on electrophysiology method accuracy (i.e.
signal to noise), and accuracy in the modelling is dependent on accuracy of input criteria.
Although eCAP recording may be limited in certain aspects and is highly sensitive to fibre
diameter-related velocity, it's still a highly used approach to evaluate nerve engagement.
We have added this point on Page 13. There are further other mechanistic considerations to
this point:

- a. Charge conduction block: It has been demonstrated in the referenced publication
that unmyelinated axons are usually blocked at frequencies above 4KHz [*Thai 2005-*
*Tai C, de Groat WC, Roppolo JR. Simulation analysis of conduction block in*
*unmyelinated axons induced by high-frequency biphasic electrical currents. IEEE*
*Trans Biomed Eng. 2005 Jul;52(7):1323-32. doi: 10.1109/tbme.2005.847561. PMID:*
*16041996; PMCID: PMC2820275*]. Therefore, this factor may have minimum impact
on eCAP amplitude in this work.
 - b. In terms of dispersion effect, the distance between site of stimulation, and the point
of recording is less than <3cm. Therefore, dispersion effects should have a minimal
impact on the peak of the eCAP [Ref- Freeman WJJ 1972 Spatial divergence and
temporal dispersion in primary olfactory nerve of cat J. Neurophysiology. 35 733–
44]. We have also quite well studied conduction velocity slowing and speeding
phenomena for the splenic nerve. The data will be published in a separate
manuscript where different frequencies are used (mainly because of the volume of
data generated). We can include a discussion on this aspect in the main manuscript
if the reviewer wishes in the next round.

In conclusion, for the estimations in this paper, we assume 100% recruitment but that may
not be the case in reality.

- 2. In relation to the second point of the reviewer's comment, i.e. how compound action
potential relates to therapeutic efficacy is an important question. We acknowledge this
limitation on Page 14, first paragraph. The present manuscript is mainly set out to answer
the question of stimulation parameters, and the relation of compound action potential vs
therapeutic efficacy is being drafted as a separate manuscript. We have also added the
reference to the clinical trial, which has recently started, and will be published as another
manuscript. A part of this question has also been addressed in the attached manuscript for
reviewer's reference which is being reviewed in PNAS. The work provides evidence for
therapy potential for immunomodulation and suppression of inflammation in large animal
model. Additional data from chronic porcine neuromodulation studies are also prepared in
another manuscript in support of this and are in line with the acute data.

We hope the reviewer find our responses in line with his/her expectations. We will be
happy to address any further questions.

Reviewer #2 (Remarks to the Author):

Gupta et al utilized computational modeling and ex vivo electrophysiology studies to identify
the optimal parameters for stimulation of the splenic neurovascular bundle (SNVB). Their
approach is thorough and builds on multiple similar previous efforts which have used a hybrid
approach to designing electrical stimulation therapies. Their findings are in line with the
existing literature on electrical stimulation and the specific parameters that dictate activation
thresholds and recruitment curves.

We thank the reviewer for acknowledging the work.

I have a few follow up questions.

1. The study utilizes complex geometries to determine the optimal stimulation parameters and
reports mostly on activation thresholds and recruitment curves. What about charge density at
the other tissues in the SNVB during optimal stimulation?

Thank you for this comment. All tissues in this application will experience a charge-balanced
pulse shape, therefore it is not expected to observe any charge accumulation in the tissues.
However, the situation can be slightly different for blood. Due to its flow, only partial volumes
may experience full charge balance. Even in the case of blood there should be no damage,
adverse effects, or safety concerns unless the cells are heated [Ref: Black, D. R., and L. N.
Heynick. "Radiofrequency (RF) effects on blood cells, cardiac, endocrine, and immunological
functions." *Bioelectromagnetics* (2003): S187-95]. We have also performed pre-clinical safety
and GLP studies that have specifically looked into damage/effects in long-term stimulation. We
have not found any evidence of damage.

2. In line with #1, one of the primary motives of the study was to understand safety limits,
however there is no discussion or mention of how the identified results fit into the current and
widely accepted definitions of safety limits for electrical stimulation.

We thank the reviewer for this comment. We have now discussed the safety aspects in
Discussion section on Page 14. As it is widely known from the literature, nerve damage may
result from electric fields or electrochemical processes at the nerve-electrode interface,
depending on many parameters including charge density (Cogan et al. 2016).
Histomorphometry and electrode design considerations suggest that nerve stimulation levels
determined in this study can be safely applied to the SNVB.

For example, distance between the electrode surface and target nerves suggests that damage
due to charge density is unlikely to occur. Near-field stimulation of less than 50 μm can
increase the likelihood of nerve damage due to nonuniform current distribution at electrode
edges (McIntyre and Grill 2001; Shannon 1992). Beyond this, charge density becomes

increasingly uniform in the mid-field and far-field range. A majority of the nerve fibres in
the SNVB are in the mid- to far-field location, particularly after fibrotic encapsulation of the
implanted device, significantly reducing the likelihood of nerve damage.

Device design may also reduce nerve damage likelihood by considering the electrochemical
limits of electrode materials. Standard smooth platinum iridium electrodes have a known
charge injection limit of 50-150uC/phase/cm² (*Rose and Robblee 1990*) that would limit nerve
recruitment and therapeutic potential. High surface area treatments of platinum iridium and
electrode coatings with higher charge injection limit materials, increase the limit for stimulation
into the mC range enabling safer transfer to nerves in the SNVB and facilitate device
miniaturization (*Park, Takmakov, and Lee 2018; Cogan 2008; Won et al. 2018*).

**3. The electrode in this study is designed to fit over the splenic artery and the surrounding fat
and nerves. In clinical applications, this electrode would be stimulating a structure that is
beating and has active blood flow. How might these variations in the state of the tissue impact
the efficacy of the stimulation?**

Thank you for an excellent question. From the perspective of this manuscript, and the
theoretical study described in this manuscript, we have not analysed the pulsatile nature of the
artery and how that might impact the efficacy of stimulation. We have acknowledged this in the
Discussion section on Page 15. We aim to perform this modelling study in future.
Computational tools can be used to morph the bundle geometry by expanding and reducing the
vessel diameter while keeping the volume of other tissues in place and with the fixed electrode
geometry.

Furthermore, we have partly studied this aspect in detail in pig chronic studies. We have seen in
preliminary data, that in the chronically encapsulated setting, there is minimum to no effect of
the dynamics of arterial pulsation on the used cuff which will be covered in future publications.
This might correlate to minimum impact on efficacy of stimulation; however, this cannot be
concluded yet.

**4. Figure 4e. What are the values for the axes?**

Thank you for the comment. We missed to add the axis. We have amended the Figure 4e to
indicate the x and y axis as Time (s) and CAP (mV) respectively.

**In limitations should also consider application in the intact in vivo condition. There it is much
more than profile of axon activation; it is also how the autonomic neural networks respond to
such stimuli. For example, using functional organ readouts, thresholds are decreased and
response curves left shift when peripheral nerves are disconnected from central aspects.
Endogenous reflexes oppose the exogenous bioelectric intervention.**

As noted by the reviewer, the translation of these stimulation parameters to in-vivo condition
can account for much broader parameters. We acknowledge this as a limitation of ex-vivo/in-

silico work on Page 13/14 (last paragraph). The impact on stimulation parameters *in-vivo* is
going to be presented in future publications.

*****

**We hope the reviewers find our response satisfactory. We welcome any further questions.**

**On behalf of all the co-authors,**

**Thank you,**

**Isha Gupta**

REVIEWERS' COMMENTS:

Reviewer #1 (Remarks to the Author):

My comments have been addressed. Nothing further to add.

Reviewer #2 (Remarks to the Author):

No further comment - congrats on nice study